# Closing the HIV Treatment Gap for Adolescents in Windhoek, Namibia: A Retrospective Analysis of Predictors of Viral Non-Suppression

**DOI:** 10.3390/ijerph192214710

**Published:** 2022-11-09

**Authors:** Farai Kevin Munyayi, Brian van Wyk

**Affiliations:** School of Public Health, University of the Western Cape, Cape Town 7535, South Africa

**Keywords:** adolescents, HIV, antiretroviral, viral suppression, viral non-suppression, adherence

## Abstract

Windhoek joined the Fast-Track Cities Initiative in 2017 to optimize HIV service delivery for adolescents, promoting adherence and sustaining viral suppression. Recent surveys and programmatic data show that the treatment gap remains greatest among children and adolescents living with HIV. A retrospective cohort analysis of adolescents living with HIV (ALHIV) receiving antiretroviral therapy (ART) at Windhoek healthcare facilities was conducted. Routine clinical data were extracted from the electronic Patient Monitoring System (ePMS). The SPSS statistical package was used to determine viral non-suppression and perform inferential statistics. 695 ALHIV were analysed with median age of 16 years (IQR = 13–18). Viral non-suppression at 1000 copies/mL threshold was 12%. Viral non-suppression was associated with age at ART initiation, duration on ART, current ART regimen and WHO Clinical Stage. In multivariate analysis, longer duration on ART was a protective factor for viral non-suppression (13–24 months vs. >24 months: aOR = 8.92, 95% CI 2.60–30.61), while being on third line regimen (vs. first line) was protective against viral non-suppression (aOR = 0.11, 95% CI 0.03–0.49). A significant treatment gap is evident for ALHIV with high viral non-suppression levels. Interventions are required to counter treatment fatigue to keep adolescents engaged in ART, and timely switching to rescue regimens for failing adolescents.

## 1. Introduction

The Fast-Track Cities Initiative, which was launched at the occasion of the 2014 World AIDS day, positioned cities as a critical component playing an essential role in the HIV response worldwide [1]. The initiative recognizes the multifaceted factors that make urban areas—with high migration rates, unemployment, and social and economic inequalities—riskier for HIV transmission than rural areas [2]. On the other hand, cities are also centers of education, economic growth, positive social change, innovation, and sustainable development, and thus provide opportunities to accelerate the HIV response towards achieving epidemic control by 2030 [1]. Windhoek is one of the Fast-Track cities that emphasizes the need to optimize HIV service delivery for adolescents, promoting adherence, enhancing retention and long-term engagement in HIV care, and maintaining sustained viral suppression [2].

Adolescents present an increasing population of people living with HIV (PLWHIV) across the world, with an estimated 1.75 million (1.16–2.3 million) adolescents living with HIV (ALHIV) in 2020 [3]. The World Health Organization (WHO) describes adolescence as the life phase between childhood and adulthood and defines adolescents as individuals aged between 10 years and 19 years [4]. Globally, in 2020 alone, an estimated 150,000 adolescents aged 10–19 years were newly infected with HIV, and these newly infected adolescents add to the already growing population of perinatally infected children who are reaching adolescence [3]. The 2021 UNAIDS global AIDS update reported that PLWHIV are facing a double jeopardy of HIV and COVID-19, while children continue to be underserved regarding access to HIV services [5]. If the current trends continue unabated, turning the tide on ALHIV will continue to be the biggest challenge to achieving HIV epidemic control. More concentrated, focused efforts are required on adolescents and young people living with HIV to make significant progress towards achieving epidemic control [3]. The UNAIDS, UNICEF, and other international partners launched “ALL in! to End Adolescent AIDS” in 2015, a global initiative with targets for 2020 to end AIDS among adolescents by 2030 [3]. It is imperative for global, regional, and national level programs to regularly monitor progress towards these targets and redesign and implement evidence-based interventions for adolescents as needed.

According to the Namibia Population-based HIV Impact Assessment (NAMPHIA), there were an estimated 11,057 ALHIV in Namibia in 2017, with an HIV prevalence of 3.7% in the older adolescents (15–19 years) and 1.9% prevalence in the younger 10–14 years age group [6]. The report also showed that the highest annual HIV incidence was among adolescent and young women aged 15–24 years (0.99%) compared to 0.36% for all adults aged 15–64 years [6]. Owing to the age disaggregation of routine data by ages of <15 years old as children and 15 years and above as adults, viral suppression estimates for the adolescent sub-population aged 10–19 years are difficult to obtain. A recent study reported viral suppression of 87% (68% fully suppressed (<40 copies/mL) and 19% low-level viremia (40–999 copies/mL)) for adolescents receiving ART at a referral hospital in Windhoek [7].

Globally, considerable advancements have been made in the development of biomedical interventions for individuals living with HIV and in expanding access to antiretroviral therapy (ART) [8]. The main goal for administering HIV treatment is to achieve undetectable viral loads, which consequently improves the health of HIV-infected individuals by facilitating immune system restoration, reducing AIDS-related illnesses and death, as well as reducing the risk of onward transmission of HIV to others [9,10,11]. Quantification of plasma viral loads is an integral component of bio-clinical monitoring of individuals who are infected with HIV and receiving ART. Viral-load monitoring provides an authentic mechanism for early detection of treatment failure and may identify challenges with adherence to treatment, detect the development of resistant mutations, and alert clinicians on timely switching to second-line ART regimens [12,13]. The World Health Organization (WHO) recommended viral load monitoring as the gold standard mechanism for keeping track of the effectiveness of HIV treatment in 2013 [14].

Adolescents living with HIV and receiving ART reportedly have an increased likelihood of poorer treatment adherence and viral suppression compared to adults receiving ART, who achieve higher rates of viral suppression [15,16,17]. Contributing to these treatment outcome disparities are the unique challenges adolescents face with taking their medication because of several factors related to their age, developmental stage (rapid physical, physiological, and psychological changes), behavior, lack of disclosure, stigma, psychosocial support, childhood forgetfulness, and caregiver type and support, among other factors [15,18]. Socioeconomic status and accessibility to adolescent-friendly health services are also attributed to the disparity in treatment outcomes between children and adolescents living with HIV (CALHIV) and adults [19]. In recent years, the COVID-19 pandemic has resulted in unintended disruptions to HIV services with many adolescents being out of school and disruptions to school-based support systems for school-going adolescents on treatment [5]. Monitoring of treatment outcomes among adolescents becomes even more essential as they become conscious of their HIV status and become more independent in managing their adherence to treatment, clinic attendance, staying engaged in care, and navigating the healthcare system [20]. Whilst younger adolescents (aged 10–14 years) are predominantly those who were infected perinatally and are more dependent on caregiver support, the older adolescents aged 15–19 years would assume increased responsibility for their own HIV care, presenting diverse challenges within this heterogenous population [21]. Routine data to monitor the adolescent age group are often not available due to routine reporting of disaggregated data of children as below 15 years old and adults as 15 years and above, which regrettably masks the monitoring of the HIV treatment outcomes, specifically for the adolescent sub-population [16,18]. This paper reports on the clinical and demographic profile of the ALHIV who are registered to receive ART in the Windhoek district and identify potential determinants of unsuppressed viral loads among ALHIV in Windhoek.

The Windhoek district is the only district in the Khomas region of Namibia. The Khomas region hosts the capital city, Windhoek, and is located in the central part of Namibia, home to an estimated 431,000 residents. According to the 2017 NAMPHIA report, Namibia had achieved 86–96–91% of the 90–90–90 UNAIDS and WHO targets with an overall 4.0% prevalence of HIV among young people (15–24 years) and 1.7% among young adolescents (10–14 years) [6]. The NAMPHIA report estimated the overall viral suppression at 85.4% and 75.4% for young people and younger adolescents, respectively, with an estimated 8.3% HIV prevalence and 73.6% overall viral suppression in Windhoek [6]. The most recent spectrum estimates reported that Namibia has reached 94–97–93% of the revised 95–95–95 targets. Windhoek is home to the largest number of ALHIV and adolescents currently receive ART at the two main hospitals, two health centers, and nine primary healthcare clinics that form the public healthcare facilities infrastructure for the district. All ART facilities included in this study operate as outpatient service delivery points, although one of the hospitals has a specialized pediatric HIV clinic.

## 2. Materials and Methods

A quantitative, retrospective cohort analysis was conducted of routine clinical data of adolescents aged 10–19 years registered at the 13 public healthcare facilities in the Windhoek district between January 2019 and December 2021. We extracted individual patient-level data for all ALHIV from the standard routine patient-monitoring data system used at all public healthcare facilities in Namibia, the electronic Patient Monitoring System (ePMS). Data extraction was conducted on the 25th of February 2022, and it consisted of anonymized data with unique patient identifiers, sociodemographic variables, and the linked clinical and treatment outcomes data recorded at routine clinic visits. Facility-based data clerks enter patient data into the ePMS from all clinic visits and the services provided during the visit as documented by clinicians in the patient files, the Patient Care Booklets (PCB). However, the electronic system only provides the last viral load test conducted on the line list report. Patient Care Booklets for patients with incomplete records in the electronic database were retrieved, and the missing information was updated in the electronic database.

The primary focus of this study was viral load non-suppression—which was defined as a viral load ≥ 1000 RNA copies/mL after at least six months of HIV treatment as recommended by WHO [22]. The Namibia HIV program further distinguishes individuals with viral loads < 40 RNA copies/mL as fully suppressed, whilst those with viral loads between 40–999 RNA copies/mL are classified as having low-level viremia. According to the Namibia antiretroviral treatment guidelines, the first viral load is conducted at 6 months after ART initiation, and at 6 monthly intervals for children < 19 years whilst individuals aged 19 years and older will have viral load testing conducted once every 12 months. Individuals with high viral loads will have more frequent viral load testing as required [23]. Antiretroviral therapy is provided free of charge and an initial adherence counseling session is done with the adolescent and/or a parent, guardian, or caregiver as appropriate at ART initiation. We utilized viral suppression in this study, as several ART adherence studies have used viral load as a proxy and standard biomarker of adherence levels and showed that it is a reliable predictor of good ART adherence [24]. However, in Namibia, healthcare workers routinely assess ART adherence through self-reporting and pill counts during clinic visits. Predictor variables extracted from the database included sex, age, age at ART initiation, WHO stage at ART initiation, ART regimen at initiation, duration on ART, current ART regimen, HIV disclosure status, TB screening results, retention in HIV care status, and level of the healthcare facility.

Extracted data were merged into a Microsoft Excel (Microsoft Corporation, Washington, DC, USA) spreadsheet, matching patient data from the ePMS and PCBs using the unique ART numbers allocated as unique patient identifiers at each clinic. The complete extracted anonymized data was saved into a password-protected Excel file to ensure that the data could not be altered. The complete Excel spreadsheet was imported into the SPSS statistical software (IBM Corp. Released 2019. IBM SPSS Statistics for Windows, Version 26.0. Armonk, NY, USA, IBM Corp.) for analysis. Descriptive statistics were performed to describe the clinical and demographic characteristics of the adolescents included in the study. Bivariate analysis was performed using the Chi-square test to determine the significance of associations between viral non-suppression and the demographic and clinical variables (sex, age, age at ART initiation, WHO stage at ART initiation, ART regimen at initiation, duration on ART, current ART regimen, HIV disclosure status, TB screening results, retention status, and level of healthcare facility). The variables which were significant at the 10% level were fed into the multivariate logistic regression model to determine risk factors for viral non-suppression. Fisher’s exact test was used as an alternative to the Chi-square test in instances of sparse data (<5 in any cell). A multivariate logistic regression model was performed, and odds ratios were used to determine predictors of viral non-suppression among adolescents on ART, with the significance level of association set at *p* < 0.05.

The ethical clearance was obtained from the University of the Western Cape Biomedical Research Ethics Committee (ref. no. BM21/5/7) in Cape Town, South Africa, and the Namibia Research Management Committee (ref. no. 17/3/3/FKM) based at the Ministry of Health and Social Services (MoHSS). We also obtained permission from the MoHSS to access the Patient Care Booklets (PCB) and electronic databases. To ensure confidentiality and respect for the privacy and dignity of the participants, no personal identifying information such as patient name/surname or identity number was extracted from the database or during the review of individual PCBs. This study was conducted in compliance with the 1964 Declaration of Helsinki guidelines and its subsequent amendments.

## 3. Results

A total of 695 adolescents aged 10–19 years accessing ART services between January 2019 and December 2021 at the 13 Windhoek district facilities were included in this analysis. The total number of ALHIV in this study was all-inclusive of all adolescents receiving ART in Windhoek district during the study period. The median age of the participants was 16 years (IQR = 13–18). The majority of adolescents included in this study were older adolescents (15–19 years) (62%), female (56.7%), had started taking ART between the ages of 0–9 years (73.5%), and were on ART for more than 24 months (85%). Namibia adopted the test-and-treat approach, also known as treat-all, whereby all individuals diagnosed HIV-positive are immediately initiated on ART [23]. Approximately 92% of the adolescents were initiated on a non-nucleoside reverse transcriptase inhibitor (NNRTI)-based regimen and 93% have remained on a first-line ART regimen. Among the 10–19 years age group, 99% of the adolescents were aware of their HIV status, having gone through the HIV disclosure process or having been diagnosed at an age that permits disclosure of HIV testing results. Most adolescents (67.3%) were classified as WHO clinical stage 1 at ART initiation, and only 2.6% screened positive for TB. The median duration of being on ART was 106 months (IQR 55–138 months). The majority of adolescents included in our study were classified as WHO clinical stage 1 at ART initiation, and approximately 30% of the adolescents constituted WHO clinical stages 2, 3, and 4. Overall, viral non-suppression at 1000 copies/mL threshold was at 12% (unsuppressed with a VL ≥ 1000 copies/mL = 12%, suppressed with VL between 40–999 copies/mL = 13.6%, and fully suppressed with VL < 40 copies/mL = 74.4%).

The results of the bivariate analysis of viral non-suppression (≥1000 copies/mL) by demographic and clinical characteristics of the adolescents on ART in Windhoek are summarized in Table 1. At a 10% level of significance, viral non-suppression status was associated with age at ART initiation (*p* = 0.079); duration on ART (*p* < 0.001); current ART regimen (*p* < 0.001); and WHO clinical stage at ART initiation (*p* < 0.001). There were no significant associations between viral non-suppression status and sex, age-group, ART regimen at initiation, HIV disclosure status, TB screening results, type of facility, and retention in care (*p* > 0.10).

After controlling for possible confounders, unsuppressed viral loads were significantly associated with duration on ART and the current ART regimen (Table 2). Patients who were on ART for 13–24 months were approximately nine times more likely to have unsuppressed viral loads compared to those who had been on ART for more than 24 months (adjusted Odds Ratio (aOR) = 8.922, 95% CI 2.601–30.611). Regarding the current ART regimen, patients who were on third-line treatment (aOR = 0.113, 95% CI 0.026–0.494) were less likely to have their viral loads unsuppressed compared to their counterparts on first-line treatment. Viral non-suppression was not significantly influenced by the age at ART initiation or the WHO clinical stage at ART initiation (*p* > 0.05) when accounting for the duration on ART and the current ART regimen.

## 4. Discussion

Our study reports viral non-suppression levels among adolescents at 12%, which is comparable to what a previous study in Windhoek reported (viral suppression = 87%) [7] and slightly higher suppression levels compared to what was reported in South Africa (85% and 80%) and Eswatini (84%) [24,25,26]. However, the viral suppression levels among adolescents still falls short of the UNAIDS and WHO target for viral suppression of 95%. The persistence of the treatment gap across various settings calls for the development of new innovations that can be more effective in addressing the achievement and maintenance of viral suppression and facilitating re-suppression among adolescents. Evaluations of the current Enhanced Adherence Counseling (EAC) intervention may provide insights into some of the adherence challenges contributing to the treatment gap and improve the implementation of EAC. The introduction of rescue regimens also comes into play where viral non-suppression persists with good adherence and the possible presence of drug resistance mutations.

Adolescents who had been on ART for longer durations were less likely to be unsuppressed than those who had been on treatment for a shorter duration. Similar findings were reported in Kenya where adolescents who were on treatment for more than ten years were 1.75 times more likely to have suppressed viral loads compared to those on treatment for less than two years, although the association was not statistically significant [25]. This is consistent with the argument that individuals on treatment for shorter periods have higher chances of experiencing virological failure compared to treatment-experienced individuals, with a study conducted in Kenya reporting higher rates of viral non-suppression in adolescents on ART for 6–12 months compared to those on ART for longer periods [13]. Treatment-experienced adolescents may benefit from the completion of the disclosure process and may have fewer psychosocial barriers and concerns such as stigma and also have improved self-competency and self-efficacy to adhere to their medication [13,27].

It is apparent from the findings of this study that the majority of adolescents were “treatment experienced adolescents” in the ‘older adolescents’ category, with most of them having been initiated on ART at an age of less than 9 years and the majority having been on treatment for a median duration of approximately 8.8 years. The higher percentage of older adolescents in this cohort may indicate the successes of the HIV program and service delivery in Namibia, i.e., the implementation of more effective treatment options and strengthening of the elimination of Mother-to-Child Transmission (eMTCT) program in Namibia, which may be resulting in substantial decreases in perinatally acquired infections. In addition, many perinatally infected adolescents are growing into older adolescents. The NAMPHIA study also reported a comparatively higher incidence of HIV in adolescents and young girls compared to the rest of the population, which could also be contributing to the higher proportions of older adolescents living with HIV [6].

Our study showed significant differences in proportions of unsuppressed viral loads according to the age of the adolescents at the time they were initiated on ART. Adolescents who were aged 10–14 years and 0–9 years when they started ART had comparatively higher levels of non-suppression (18.9% and 11.2%, respectively) than adolescents who were initiated on ART at ages 15–19 years (8.3%). Similar findings were reported in Kenya where adolescents who were diagnosed and initiated on ART at age 10–14 years had higher proportions of viral non-suppression compared to the adolescents who started ART at 15–19 years of age (27.6% vs. 9.6%) [21]. A study conducted in South Africa also provided supporting evidence whereby adolescents who were initiated at age 15–19 years were 1.4 times more likely to have a suppressed viral load whilst those initiated at 10–14 years of age were 20% less likely to be suppressed than the adolescents initiated at 0–9 years of age [10]. Older age at ART initiation has also been found to be associated with viral suppression in studies conducted in Zimbabwe and Tanzania [27,28].

The type of regimen (e.g., NNRTI) for children and younger adolescents, combined with transmitted resistance and possible acquired drug resistance from factors associated with appropriate dose adjustments by weight in children and changes in the metabolism of medication with age, have been associated with higher odds of virologic failure in younger population subgroups [27,28]. Poor virologic suppression in younger populations has also been associated with broader transition challenges to adult care, which includes managing disclosure. At initiation, older adolescents may acquire better self-efficacy and self-competency for ART adherence due to their higher intellectual maturity and understanding to better understand adherence counseling and take responsibility for the self-management of their chronic condition [13]. However, existing evidence reported in a systematic review and meta-analysis on the efficacy of interventions for self-management to improve health-related outcomes in ALHIV was inconclusive, and these interventions may heavily rely on “the individual, social and health system contexts” [29,30].

As expected, the majority of adolescents on ART were initiated on an NNRTI-based regimen as prescribed in the National ART Guidelines. The proportion of adolescents with viral non-suppression was highest among adolescents who were initiated on an NNRTI-based regimen compared to a protease inhibitor (PI)- or dolutegravir (DTG)-based regimen, but the differences did not significantly affect viral suppression. This finding can be attributed to the very small numbers of adolescents started on non-NNRTI-based regimens. It is also interesting to note that none of the eleven adolescents who were initiated on a DTG-based regimen were virally unsuppressed and only three of forty-eight adolescents who started on a PI-based regimen were unsuppressed. The WHO recommended DTG as the preferred HIV treatment option due to having greater effectiveness, fewer side effects, and a high genetic barrier to developing drug resistance [31]. A large cohort study on DTG conducted in 2019 also showed excellent efficacy as a first-line or subsequent switch regimens, supporting the utility of transitioning unsuppressed adolescents to DTG-based regimens [30].

Most of the adolescents remained on a first-line regimen and achieved and maintained viral suppression with a 9.7% proportion of non-suppression, whilst those who were switched to a second-line regimen had a higher proportion of viral non-suppression at 46.3%. However, neither of the two adolescents currently on a third-line regimen were virally unsuppressed and there was no significant difference in viral non-suppression between adolescents on second-line regimens compared to those on a first-line regimen. On the other hand, findings from two South African studies reported higher viral non-suppression levels among adolescents on second-line ART compared to those on first-line regimens [10,24]. A study conducted in Tanzania found non-adherence to a first-line regimen as a predictor of non-adherence to second-line ART regimens [32], and likewise, a Ugandan study showed evidence that adolescents with a history of treatment failure were at higher risk of viral non-suppression even with regimen switches [16]. Further exploration into persisting viral non-suppression among adolescents, and the dynamics between EAC and other interventions, through regimen switches may provide insights into some of the barriers unsuppressed adolescents are experiencing that are non-biomedical in nature.

This study was a retrospective cohort study that relied on routinely collected patient data during clinical interactions at clinic visits. The completeness and accuracy of the available data were dependent on the data collection and capturing of patient records by the clinicians and data clerks; this meant that we could not control the selection of variables of interest outside of those routinely collected and could not account for other unmeasured clinical and socioeconomic factors as covariates. Other variables, such as CD4 count, alcohol use, and pregnancy status for female adolescents, were poorly and inconsistently captured and could not be reliably included in the analyses. The amount of missing HIV disclosure, TB screening, WHO clinical stage at ART initiation, and viral load data could have potentially affected the findings of this study. The ePMS has limitations in obtaining longitudinal data and only the last viral load results could be obtained from the system; thus, we could not obtain consecutive viral load results, so virally unsuppressed adolescents may have had viral blips. On the other hand, the analysis of routinely collected clinical data provides a real-life lens into the HIV service delivery’s strengths and gaps for the study population, including the information gaps, as reflected by the incomplete data. Our study sample was also all-inclusive of all adolescents, reducing biases such as sampling or selection bias.

## 5. Conclusions

Our study demonstrates comparatively high rates of viral non-suppression among the adolescent population living with HIV in Windhoek, Namibia. Viral suppression is close to the previous 90% target but requires reinvigoration to reach the new 95% threshold. The remaining gap towards the achievement of the revised UNAIDS and WHO targets for epidemic control calls for urgent attention to the implementation of targeted interventions to improve adherence to treatment among adolescents. The study also confirms that the longer we can keep adolescents engaged in treatment, the lesser risk of viral non-suppression. Regular viral load monitoring is essential to identify virologic failure and switch failing adolescents to more potent rescue regimens appropriately. Potentially perinatally infected ALHIV who enter later adolescence, which coincides with the transition to adult care, are at the greatest risk for viral non-suppression, and thus require intensive intervention to combat treatment fatigue as well as psychosocial issues emanating from long-term adherence requirements and lifestyle and development changes. Leveraging successful evidence-based strategies and new innovations will be key in closing the treatment gap among ALHIV and can potentially reduce rates of viral non-suppression.

## Figures and Tables

**Table 1 ijerph-19-14710-t001:** Viral non-suppression status by demographic and clinical characteristics of adolescents (10–19 years) on ART in Windhoek District, Namibia: 2019–2021 (*n* = 695).

Characteristic	Total		Unsuppressed Viral Load		Chi-Square Statistic	*p*-Value
	** *n* **	**%**	** *n* **	**%**		
**Sex**						
Female	394	56.7	47	13.0	0.771	0.380
Male	301	43.3	31	10.8		
**Age group (years)**						
10–14	264	38.0	30	11.7	0.048	0.827
15–19	431	62.0	48	12.2		
**Age at ART initiation (years)**						
0–9	511	73.5	56	11.2	5.087	**0.079 ***
10–14	96	13.8	17	18.9		
15–19	88	12.7	5	8.3		
**Duration on ART (months)**						
0–12	12	12.2	5	9.8	24.750	**<0.001 ***
13–18	10	1.4	4	40.0		
19–24	10	1.4	5	55.6		
>24	591	85.0	64	11.1		
**ART regimen at initiation**						
NNRTI based	636	91.5	75	12.6	2.411	0.299
PI based	48	6.6	3	6.4		
DTG based	11	1.6	0	0.0		
**Current ART regimen**						
First line	647	93.1	59	9.7	48.938	**<0.001 ***
Second line	46	6.6	19	46.3		
Third line	2	0.3	0	0.0		
**HIV Disclosure status**						
Disclosed	667	99.0	74	11.8	0.403	0.686
Not disclosed	7	1.0	0	0.0		
**TB Screen results**						
Incomplete/missing	23	3.3	2	9.5	0.130	0.937
Negative	654	94.1	74	12.1		
Positive	18	2.6	2	12.5		
**WHO clinical stage at ART initiation**						
1	378	67.3	28	8.0	22.416	**<0.001 ***
2	100	17.8	23	25.3		
3	65	11.6	12	19.0		
4	19	3.4	3	17.6		
**Facility level**						
Health centre	175	25.2	17	10.5	3.351	0.187
Hospital	343	49.4	37	10.9		
Primary health clinic	177	25.5	24	16.3		
**Retention in care**						
Retained	588	84.6	66	11.5	1.520	0.249
Not retained	107	15.4	12	16.4		

* 10% level of significance (*p* < 0.1).

**Table 2 ijerph-19-14710-t002:** Determinants of viral non-suppression among adolescents (10–19 years) on ART in Windhoek District, Khomas Region, Namibia 2019–2021 (*n* = 695).

Characteristic	Unadjusted Estimates				Adjusted Estimates			
	Odds Ratio	95% Confidence Interval		*p*-Value	Odds Ratio	95% Confidence Interval		*p*-Value
**Age at ART initiation (years)**								
0–9	0.126	0.096	0.167	**<0.001 ***	1.543	0.404	5.889	0.525
10–14	0.233	0.137	0.395	**<0.001 ***	2.119	0.594	7.556	0.247
15–19 (Ref.)	1.000							
**Duration on ART (months)**								
0–12	0.100	0.040	0.251	**<0.001 ***	1.277	0.306	5.319	0.737
13–24	0.900	0.366	2.215	**0.819**	8.922	2.601	30.611	**0.001 ***
>24 (Ref.)	1.000							
**Current ART regimen**								
Third Line	0.108	0.082	0.141	**<0.001 ***	0.113	0.026	0.494	**0.004 ***
Second line	0.864	0.467	1.596	0.640	0.825	0.164	4.154	0.815
First Line (Ref.)	1.000							
**WHO clinical stage at ART initiation**								
1	0.088	0.059	0.129	**<0.001 ***	0.319	0.091	1.117	0.074
2	0.338	0.211	0.543	**<0.001 ***	1.309	0.352	4.875	0.688
3	0.235	0.125	0.441	**<0.001 ***	0.877	0.217	3.545	0.854
4 (Ref.)	1.000							

* 5% significance level (*p* < 0.05).

## Data Availability

The dataset extracted and analyzed in this study are not publicly available as the Namibia Ministry of Health and Social Services is the custodian of all routine patient-level data. Data are available from the corresponding author on reasonable request.

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
