# Peer review of "Closing the HIV Treatment Gap for Adolescents in Windhoek, Namibia: A Retrospective Analysis of Predictors of Viral Non-Suppression"

_ijerph, 2022, doi:10.3390/ijerph192214710_

Round 1

Reviewer 1 Report

The manuscript entitled “Closing the HIV treatment gap for adolescents in Windhoek, Namibia: a retrospective analysis of predictors of viral non-suppression” by Munyayi and van Wyk is an interesting paper adding supporting new data about antiretroviral therapy (ART) outcomes among adolescents living with HIV (ALHIV) in a district of Namibia. To this aim, Authors conducted a retrospective cohort analysis of predictors and risk factors of viral non-suppression in 695 ALHIV receiving ART at Windhoek healthcare facilities by extraction of routine clinical data from the electronic Patient Monitoring System (ePMS).

The article is clear and well written, the results are well presented, the conclusions drawn are supported by the results obtained, and the findings give a potential improvement to the knowledge of the determinants of viral non-suppression among ALHIV in Namibia and to the factors contributing to the treatment gap. These interesting findings may contribute to the evaluation of new innovations to address achievement and maintenance of viral suppression among ALHIV, thus, after clarifying a minor point, the manuscript is suitable for publication in International Journal of Environmental Research and Public Health.

Minor point:

Sentence at lines 18-19 in the Abstract “In multivariate analysis, longer duration on ART was a risk factor for viral non-suppression….” seems to be at odds with the following sentences of Section 3. Results, Page 6, Lines 203-206 “Patients who  were on ART between 13 - 24 months were approximately 9 times more likely to have  unsuppressed viral loads compared to those who had been on ART for more than 24 months...”, and of Section 4. Discussion, Page 7, Lines 230-231 “Adolescents who had been on ART for longer duration were less likely to be unsuppressed than those who had been on treatment for a shorter duration.”

Please clarify.

Reviewer 2 Report

The compliance of the subjects to ART should be described, and the (eventual) methods used by health services to assure the adherence to the therapy.

The Authors should define the reprentaiveness of the 695 adolescents recruited by health services on HIV affected on the age range in the district. 

There are not differenecs among health centers, hospital and primary health clinic: the seriousness of the HIV disease is supposed to be higher in Hospital group, please discuss properly the item, even in association of previous paper of the Authros concerning teen clubs.

Factors infleuncing age initation of therapies should be described.

Moreover, a brief description of the organization of helath services distributing therapies is needed. Is the therapy free of charge (I presume)? Is the therapy accompanied by some instructions for parents or caregivers? Are all the adolescents aware about the disease, the therapy and possible adverse effects?  Are these factors possibly infleuncing the success of therapy?
